# Music in noise recognition: An EEG study of listening effort in cochlear implant users and normal hearing controls

Giulia Cartocci[1,2☯]*, Bianca Maria Serena Inguscio[3☯], Andrea Giorgi[1,2], Alessia Vozzi[2], Carlo Antonio Leone[4], Rosa Grassia[4], Walter Di Nardo[5], Tiziana Di Cesare[5], Anna Rita Fetoni[5], Francesco Freni[6], Francesco Ciodaro[6], Francesco Galletti[6], Roberto Albera[7], Andrea Canale[7], Lucia Oriella Piccioni[8], Fabio Babiloni[1,2]

1 Department of Molecular Medicine, Sapienza University of Rome, Rome, Italy, 2 BrainSigns ltd, Rome, Italy, 3 Department of Sensory Organs, Sapienza University of Rome, Rome, Italy, 4 Department of Otolaringology Head-Neck Surgery, Monaldi Hospital, Naples, Italy, 5 Institute of Otorhinolaryngology, Catholic University of Sacred Heart, Fondazione Policlinico "A Gemelli," IRCCS, Rome, Italy, 6 Department of Otorhinolaryngology, University of Messina, Messina, Italy, 7 Department of Surgical Sciences, University of Turin, Turin, Italy, 8 Department of Otolaryngology-Head and Neck Surgery, IRCCS San Raffaele Scientific Institute, Milan, Italy

☯ These authors contributed equally to this work.
* giulia.cartocci@uniroma1.it

**Data Availability Statement:** Data cannot be shared publicly because of participants to the study signed a written informed consent in which it was specified that data would not be shared with third

## Abstract

Despite the plethora of studies investigating listening effort and the amount of research concerning music perception by cochlear implant (CI) users, the investigation of the influence of background noise on music processing has never been performed. Given the typical speech in noise recognition task for the listening effort assessment, the aim of the present study was to investigate the listening effort during an emotional categorization task on musical pieces with different levels of background noise. The listening effort was investigated, in addition to participants' ratings and performances, using EEG features known to be involved in such phenomenon, that is alpha activity in parietal areas and in the left inferior frontal gyrus (IFG), that includes the Broca's area. Results showed that CI users performed worse than normal hearing (NH) controls in the recognition of the emotional content of the stimuli. Furthermore, when considering the alpha activity corresponding to the listening to signal to noise ratio (SNR) 5 and SNR10 conditions subtracted of the activity while listening to the Quiet condition—ideally removing the emotional content of the music and isolating the difficulty level due to the SNRs- CI users reported higher levels of activity in the parietal alpha and in the homologous of the left IFG in the right hemisphere (F8 EEG channel), in comparison to NH. Finally, a novel suggestion of a particular sensitivity of F8 for SNR-related listening effort in music was provided.

## Introduction

For hearing impaired persons in general, and for cochlear implant (CI) users in particular, despite technology developments, there are still two main challenges: listening to music [1, 2]

parties. However we provided a minimal data set as supporting file. In the minimal data set are reported all data employed for instance for preparing figures.

**Funding:** The study was funded to BrainSigns Ltd by Cochlear Ltd (IIR 1983). The funders had no role in study design, data collection and analysis, decision to publish, or preparation of the manuscript.

**Competing interests:** The authors have declared that no competing interests exist.

and hearing in noise [3, 4]. In fact, both such issues have been strongly investigated, but it is interesting to note that the conjunction of the two was never approached, despite its ecological value. It is indeed common experience to listen to music in the crowd such as during live-music events, religious events, or even just on buses. It is well-known from literature that CI users experience difficulties in perceiving music. This is probably due to the constraint of the CI in the transmission of the spectral information of music, and to the complexity of pitch relationships between notes, both at the basis of the perception of the melody [1, 5–7]. In fact CIs are biomedical devices that provide hearing restoration to people with severe-to-profound hearing loss by transforming auditory stimuli into a direct electrical stimulation of auditory neurons in the cochlea. Currently, music has been investigated as an "environmental factor", studying its influence during the execution of other principal activities. In fact for instance it was found to be protective against perceived effort during physical exercise [8] or can increase effort while performing engaging activities like driving [9], but what about the mental effort produced by listening to music?

The study of music perception in CI users is worthwhile since, beyond the aesthetical experience, musical pitch perception, intended as the fundamental frequency perception, is tightly related to the linguistic tone perception skill, crucial especially for the comprehension of tonal languages [10]. Furthermore, recent studies highlighted the importance of music also on human brain development, involving the acquisition of cognitive, emotional, and auditory-motor processing skills [11]. In fact it is not surprising that there is a plethora of studies concerning music and cochlear implant users [1, 5, 12, 13]. Music is a complex auditory stimulus, more complex than speech and suggested to have a role in facilitating language development [14] in hearing impaired persons, and sharing commonalities with language with respect to syntax [15] and semantics [16]. The study of music perception by CI users expanded its methodological approach from participants' rating of musical characteristics and behavioural performances about the recognition of particular musical features, like pitch [1], rhythm and interval [17], to the investigation of neurophysiological correlates of music perception and processing [12]. A general result of such studies showed deficits in music perception expressed by CI users in comparison to normal hearing (NH) controls [1, 2, 7]. Such deficits have been shown to be reflected also by different cerebral activity patterns in comparison to NH controls, that were also reported to be influenced by deafness etiology and by unilateral or bilateral CI (UCI and BCI respectively) conditions, that is the condition of having and using the CI in only one or in both ears [18, 19].

Concerning the second mentioned challenge for CI users, that is hearing in noise, as well research has been extensively done through performance assessment and more recently focusing on the listening effort related to the recognition of speech or tones in noise [20–23]. The listening effort has been defined as: "The mental exertion [effort] required to attend to, and to understand, an auditory message [listening]"[24], and later "The deliberate allocation of mental resources to overcome obstacles in goal pursuit when carrying out a [listening] task"[25]. The methods of investigation of listening effort can be divided into: i) self-report and cognitive-behavioural (e.g. questionnaires [26], rating scales [27] and dual tasks [28]) and ii) physiologically-based, focusing on central nervous system activity (electroencephalography–EEG [29–32]-, functional magnetic resonance–fMRI -[33], functional near infrared spectroscopy–fNIRS—[34]) and on peripheral nervous system activity (pupillometry [35], electrodermal activity [36]).

Concerning EEG, the assessment of listening effort has mainly been estimated through the study of the variation of alpha rhythm [25] (approximately 8–13 Hz), which was observed to decrease during active processing of language stimuli [37], and possibly involved in a "gating by inhibition" mechanism, aimed at inhibiting task-irrelevant activities in task-irrelevant

regions [38]. Specifically, the extent of alpha activity suppression has been proved to influence speech intelligibility [30, 39], according to the hypothesis of an anticipatory/preparatory role of alpha for the arrival of expected stimuli [40]. In fact, in auditory tasks alpha rhythm seems to be involved in the maintenance of the activation-inhibition balance [21], being the principal rhythm in rest conditions and desynchronizing in correspondence of the anticipation or processing of a stimulus [41]. Concerning cortical areas identified as being involved in the listening effort, the involvement of parietal areas is quite consolidated, with higher levels of alpha activities linked to more difficult audibility conditions [21, 30, 42–45], but interestingly not in correspondence with conditions characterized by an excessive difficulty for the participant [30, 43, 46]. Despite this there are, on the contrary, evidences of a decrease of such alpha activity in correspondence of more difficult hearing conditions [47, 48]. In particular, alpha increased in central-parietal [22, 49] and occipital-parietal [45] areas. Concerning the hearing impaired population, the alpha rhythm as an index of the listening effort in speech-in-noise recognition tasks employing different signal and noise directions have been studied in asymmetric hearing loss children [43], in single side deaf children [50], in adult UCI users [31, 44] and also in the comparison of different CI processors, with the purpose of identifying the tools eliciting the lower listening effort [51–53].

Broca's area, located in the inferior frontal gyrus (IFG—corresponding to BA45 and F7/F8 EEG channel according to the 10–20 international system [54]), although traditionally associated to language processing [55], through cerebral activity studies has been shown to be involved in both listening effort [56] and musical syntax processing [57]. Musical syntax is a complex rule-based information processing, linked to the expectations of the listener of sequences constituted by harmonically related chords both in musicians and non musicians, and taking place in both the hemispheres, that is not only in Broca's area in the left hemisphere, but also in its right homologous [58].

The sum of the above mentioned evidences concerning IFG and Parietal alpha activities as indices of listening effort, supported their sensitivity in highlighting differences between CI users and NH controls. This led to the present hypothesis of obtaining differences between the two groups also in the current study, involving musical stimuli with background noise.

Moreover, given the traditional concept of the higher specialization of the right hemisphere for the emotional processing [59] and the evidence that right ear cochlear implanted children present deficits in emotional prosody (i.e. emotional prosody shares with music a strong reliance on features such as pitch and rhythmicity [60]) recognition [61], it could be argued that the side of the cochlear implant could influence both neurophysiological underpinnings and behavioural performances object of investigation in the present study.

Since the more recent definition of listening effort underlines the importance of motivation and involvement of the listener in the task, in the present study it has been chosen to assign a categorization task to participants, concerning the emotional content of classical musical pieces [62]. CI users in general present emotion recognition deficits in comparison to NH population [63–65], but specifically in the present study an emotional categorization task was chosen with the aim of considering the communicational intent of music [66] as a matter of investigation in CI users, given the obvious implications on communication of their clinical condition.

The first aim of the present research was the investigation in NH and unilateral UCI participants of the occurrence of listening effort-related cerebral patterns, indexed by the modulation of alpha activity in the parietal and IFG (including the canonical Broca's area), during listening to music, and in particular their elicitation through the translation of the typical speech in noise recognition task to a "music emotion in noise recognition task". In addition, a secondary aim was the assessment of the influence of the CI side on the investigated task, with respect to

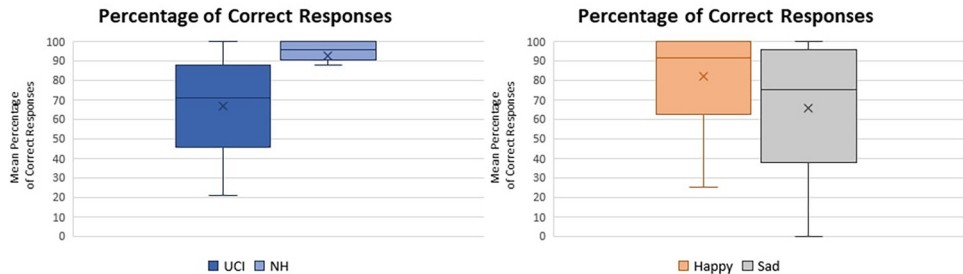

**Fig 1.** Left: box plot reporting the percentage of Correct Responses in the comparison between normal hearing (NH) and unilateral cochlear implant users (UCI). Right: box plot reporting the percentage of Correct Responses in the comparison between Happy and Sad musical pieces. Whiskers from minimum to maximum value; 25th, 50th (median) and 75th percentile drawn as horizontal lines in each box; "x" stands for the mean.

neurophysiological (EEG-based indices) and behavioural (emotion recognition performances and declared ratings) processes.

## Results

### Behavioural results

The NH group performed better recognition of the emotional content of the musical pieces in comparison to the UCI group ($F_{(1,35)} = 10.630$ p = 0.002 Partial eta-squared = 0.233) (Fig 1 left). Furthermore an effect of the emotional tone of music was shown, with happy pieces being recognized more than sad ones ($F_{(1,35)} = 7.726$ p = 0.009 Partial eta-squared = 0.181) (Fig 1 right), irrespectively of the signal to noise ratio (SNR) ($F_{(2,70)} = 0.361$ p = 0.698 Partial eta-squared = 0.010). No interaction between group, emotion and SNR variables was observed.

A further analysis conducted on the percentage of Correct Responses considering the effect of the order of exposure to the same musical piece, irrespectively of the SNR condition, showed that the order of exposure did not influence the percentage of Correct Responses ($F_{(2,68)} = 0.685$ p = 0.507 Partial eta-squared = 0.020). Moreover, there was no interaction between the factor order and the factor group (($F_{(2,68)} = 0.879$ p = 0.420 Partial eta-squared = 0.025).

Conversely to correct responses, the analysis of reaction times through ANOVA, performed considering the variables group, emotion and order, showed an effect of the order ($F_{(2,70)} = 4.599$ p = 0.013 Partial eta-squared = 0.116) (Fig 2). In particular, after the first listening

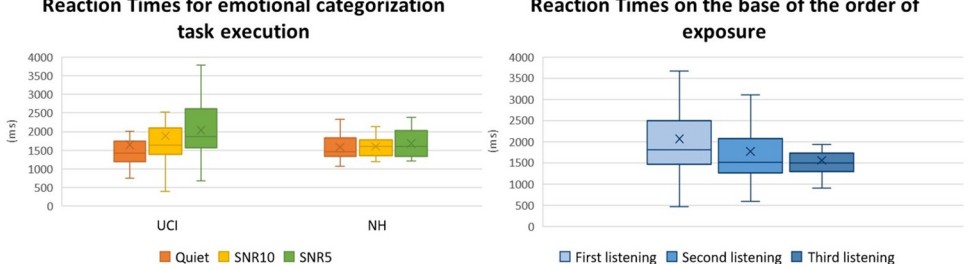

**Fig 2.** Left: Representation of the mean Reaction Times for the emotional recognition task execution of the musical excerpts, focusing on the different auditory conditions (Quiet, SNR10, SNR5). Right: Representation of the mean Reaction Time for the emotional recognition task execution of the musical excerpts, focusing on their order of appearance and irrespectively of the audibility condition (Q, SNR5, SNR10). Whiskers from minimum to maximum value; 25th, 50th (median) and 75th percentile drawn as horizontal lines in each box; "x" stands for the mean.

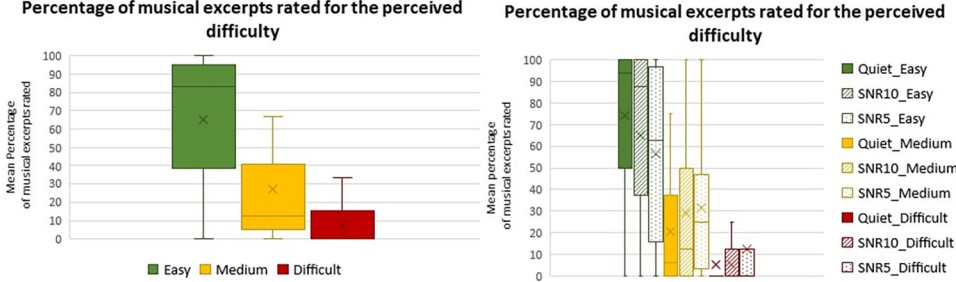

**Fig 3.** Left: box plot reporting the percentage of musical excerpts rated for the perceived difficulty: Easy, Medium, Difficult. Right: box plot reporting the percentage of musical excerpts rated for the perceived difficulty (Easy, Medium, Difficult), in combination with the different auditory conditions (Quiet, SNR 10 and SNR5). Whiskers from minimum to maximum value; 25th, 50th (median) and 75th percentile drawn as horizontal lines in each box; "x" stands for the mean.

participants exhibited lower reaction times for giving the response about the emotional categorization of musical pieces, both in comparison to the second (p = 0.036) and to the third listening (p<0.001). There was instead no difference between second and third listenings (p = 0.132). Neither the effect of the group (F(1,35) = 1.037 p = 0.315 Partial eta-squared = 0.029) or the emotion (F(1,35) = 0.090 p = 0.765 Partial eta-squared = 0.002) was observed.

Concerning the rated difficulty declared by participants, an effect of the level of difficulty rated by participants (F(2,52) = 24.133 p<0.001 Partial eta-squared = 0.481) was found. In particular, the amount of musical pieces rated as Easy to listen to was higher than the Medium and Difficult ones (both p<0.001) (Fig 3 left). There was no interaction between the variables group and difficulty rating (F(2,52) = 1.477 p = 0.238 Partial eta-squared = 0.054). In addition, an effect of the SNR (F(2,52) = 0.546 p = 0.582 Partial eta-squared = 0.020) was not found, but an interaction between SNR and difficulty rating (F(4,104) = 5.122 p<0.001 Partial eta-squared = 0.164) (Fig 3 right) was. In particular, among the musical pieces rated as Easy to listen to the ones delivered in the Quiet condition were more than the ones delivered in the SNR5 (p<0.001), vice versa among the musical pieces rated as Medium to listen to the ones delivered in the Quiet condition were less than the ones delivered in the SNR5 (p = 0.028). Additionally, there was no difference between Quiet and SNR10 and between SNR5 and SNR10, among the Easy to listen to musical pieces (p = 0.064 and p = 0.065 respectively). Similarly, there was no difference between Quiet and SNR10 and between SNR5 and SNR10, among the Medium to listen to musical pieces (p = 0.079 and p = 0.576 respectively). Among the musical pieces rated as Difficult to listen to there was no difference on the base of the SNR (all p>0.05).

On all participants, a correlation between the percentage of Correct Responses and the percentage of musical pieces rated as Easy (Pearson r = 0.525 p = 0.004) and Medium to listen (Pearson r = -0.526 p = 0.004) (Fig 6 left) was found, but not for the Difficult to listen musical pieces (Pearson r = -0.246 p = 0.207). Finally, in the UCI group the correlation between the period of CI use and the rated difficulty was investigated, showing a negative correlation with the rated Easy to listen to (Pearson r = -0.472 p = 0.048) and a positive correlation with the Medium to listen to (Pearson r = 0.469 p = 0.050) musical pieces (Fig 6 right); no correlation instead was found with the Difficult to listen to (Pearson r = 0.162 p = 0.520) musical pieces.

Concerning music listening habits, no statistically significant correlation between the declared number of hours per week spent listening to music by CI users and percentage of correct responses (Pearson r = 0.056 p = 0.855), and percentage of musical excerpts rated as Easy

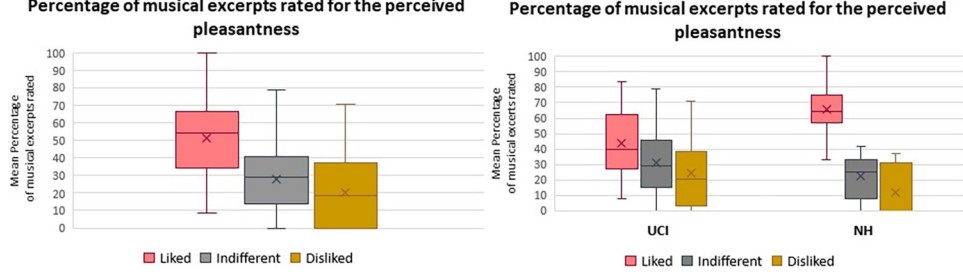

**Fig 4.** Left: box plot reporting the percentage of musical excerpts rated for the perceived pleasantness: Liked, Indifferent, Disliked. Right: box plot reporting the percentage of musical excerpts rated for the perceived pleasantness (Liked, Indifferent, Disliked), specificaly for the two Groups (UCI and NH). Whiskers from minimum to maximum value; $25^{th}$, $50^{th}$ (median) and $75^{th}$ percentile drawn as horizontal lines in each box; "x" stands for the mean.

(Pearson r = 0.155 p = 0.614), Medium (Pearson r = -0.240 p = 0.429) and Difficult (Pearson r = 0.099 p = 0.747) to listen to was found.

Concerning the perceived pleasantness rating of the musical pieces, there was an effect of the variable ($F_{(2,52)}$ = 16.423 p<0.001 Partial eta squared = 0.387) and the post hoc analysis showed that the percentage of musical pieces rated as Liked was higher than the ones rated as Indifferent (p<0.001) and Disliked (p<0.001) (Fig 4 left). Moreover there was an interaction between the factor group and the factor perceived pleasantness ($F_{(2,52)}$ = 3.972 p = 0.025 Partial eta squared = 0.132), as further evidenced by post hoc analysis characterized by an increased percentage of musical excerpts rated as Liked in NH group in comparison to all the other ratings both in the NH and in the UCI group (p<0.01 for all) (Fig 4 right).

Concerning the assessment of the potential influence of the side of the CI on emotional processes, no difference between right side and left side implanted participants belonging to the UCI group were found for the following variables: percentage of Correct Responses (t = 0.067 p = 0.947), percentage of musical pieces rated as Easy to listen to (t = -0.745 p = 0.467), percentage of musical pieces rated as Medium to listen to (t = 0.291 p = 0.775), percentage of musical pieces rated as Difficult to listen to (t = 1.335 p = 0.200), in TAS-20 score (t = 0.593 p = 0.560), percentage of musical pieces rated as Liked (t = 0.783 p = 0.445) percentage of musical pieces rated as Indifferent (t = -0.974 p = 0.344), percentage of musical pieces rated as Disliked (t = 0.057 p = 0.955).

## EEG results

Concerning all the investigated areas, an effect of the emotional content of the musical pieces was found, with the happy music resulting in eliciting higher levels of alpha activity in comparison to sad music: parietal area ($F_{(1,35)}$ = 11.691 p = 0.002 Partial eta-squared = 0.250), F7 ($F_{(1,35)}$ = 4.607 p = 0.038 Partial eta-squared = 0.116) and F8 ($F_{(1,35)}$ = 21.938 p<0.001 Partial eta-squared = 0.385). Moreover, focusing on F8 activity, an effect of the audibility condition ($F_{(2,70)}$ = 4.052 p = 0.022 Partial eta-squared = 0.104) was shown, with the SNR5 condition producing lower alpha levels than both Quiet (p = 0.031) and SNR10 (p = 0.048) condition; Quiet and SNR10 (p = 0.774) conditions did not differ between each other.

Interestingly, in F7 when subtracting from the SNR5 and SNR10 condition the Quiet condition (sub Quiet), that is ideally the emotional content of the music, the effect of emotion disappeared ($F_{(1,35)}$ = 0.913 p = 0.346 Partial eta-squared = 0.025), while in the right IFG, when subtracting the alpha activity collected during the listening to the quiet condition highlighted that UCI group reported higher levels of parietal alpha activity in comparison to NH group ($F_{(1,35)}$ = 4.737 p = 0.036 Partial eta-squared = 0.119) (Fig 5 right), as well as the Parietal area (F

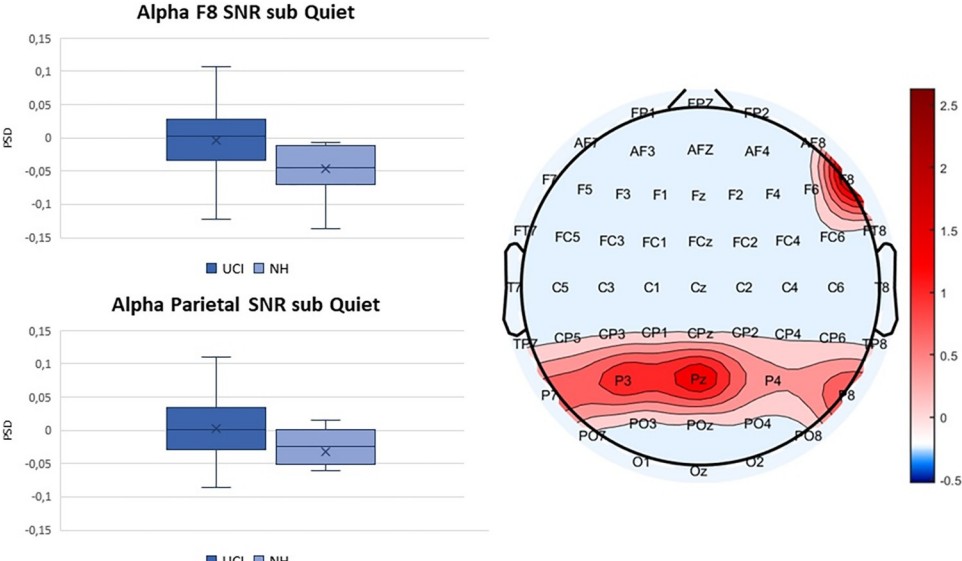

**Fig 5.** Left up: comparison between the UCI and NH groups mean values of alpha F8 activity calculated subtracting the activity recorded during the listening to the Quiet condition from the activity corresponding to the listening to the conditions characterized by the presence of background noise. Whiskers from minimum to maximum value; 25th, 50th (median) and 75th percentile drawn as horizontal lines in each box; "x" stands for the mean. Left down: comparison between the UCI and NH groups mean values of alpha parietal activity calculated subtracting the activity recorded during the listening to the Quiet condition from the activity corresponding to the listening to the conditions characterized by the presence of background noise. Whiskers from minimum to maximum value; 25th, 50th (median) and 75th percentile drawn as horizontal lines in each box; "x" stands for the mean. Right: topoplot representing the t values calculated comparing the mean alpha PSD values for each channel included into the statistical analysis reported into the graphs on the left (F8 and Pz, P3, P4, P7, P8 respectively).

$(1,35) = 4.220$ p = 0.047 Partial eta-squared = 0.107) (Fig 5 left). For all the indices subtracted of the Quiet condition there was no effect of the different SNR condition: for F7 ($F(1,35) = 0.638$ p = 0.430 Partial eta-squared = 0.018) and for F8 ($F(1,35) = 2.631$ p = 0.114 Partial eta-squared = 0.070) and for the Parietal area ($F(1,35) = 3.169$ p = 0.084 Partial eta-squared = 0.083).

Furthermore, independently of the group, the correlation was investigated between the mean alpha activity in the F7, F8 and parietal area during the conditions characterized by background noise subtracted of the activity during the quiet condition and the percentage of musical pieces rated as Easy, Medium and Difficult to listening to. The sub Quiet version of the indices was chosen as "ideally removed" of the emotional content of the musical pieces which could be more representative of the listening effort experienced by participants. Results showed a positive correlation between alpha activity estimated in F8 sub Quiet and the percentage of musical excerpts rated as Medium to listen (r = 0.381 p = 0.045) and a negative correlation between F8 sub Quiet alpha activity and the percentage of musical excerpts rated as Easy to listen (r = -0.412 p = 0.030) (Fig 6 centre), but no correlation with those rated as Difficult to listen to (r = 0.263 p = 0.176). Moreover no correlation between difficulty ratings and parietal alpha activity sub Quiet (Easy: r = -0.195; Medium: r = 0.274; Difficult r = -0.091) was found, nor with the F7 sub Quiet activity (Easy: r = -0.089; Medium: r = 0.041; Difficult: r = 0.153). Furthermore, the two groups were investigated separately for F8, given the difference between them for alpha activity SNR sub Quiet in that channel (Fig 5 Left, Up), and the correlation between alpha activity SNR sub Quiet in F8 and the percentage of difficulty rating (Easy, Medium, Difficult) was performed. Results showed that only in the UCI group it was

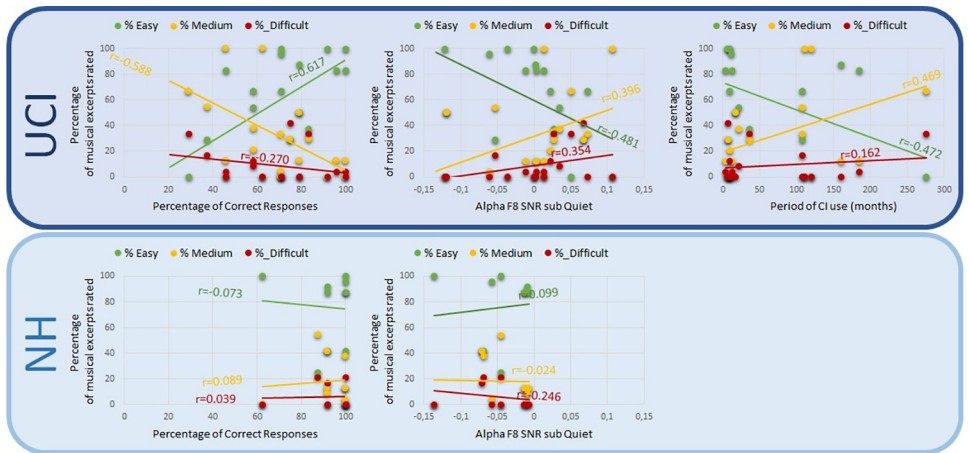

**Fig 6.** For the UCI (top row) and NH (bottom row) group graphical representation of the correlation between the percentage of musical excerpts rated as Easy (% Easy) and Medium (% Medium) to listen and: (Left) the percentage of Correct Responses reported in the emotional recognition task; (Centre) the mean alpha activity estimated in F8 channel during the conditions characterized by background noise (SNR5 and SNR10) subtracted of the activity corresponding to the listening to the Quiet condition (Q); (Right) the period of CI use in months, obviously limited to the UCI group.

maintained a negative correlation between the percentage of musical excerpts rated as Easy and the F8 SNR sub Quiet alpha activity (r = -0.481 p = 0.044) and there was a trend of a correlation between the percentage of musical excerpts rated as Medium and the F8 SNR sub Quiet alpha activity (r = 0.396 p = 0.104). In contrast, in the NH group such correlations were not found (Easy: r = 0.099 p = 0.786; Medium: r = -0.024 p = 0.948).

Furthermore, concerning the assessment of the potential influence of the side of the CI on neurophysiological correlates, no difference between right side and left side implanted participants was found belonging to the UCI group for both the mean F8 alpha sub Quiet (t = -0.049 p = 0.961) and mean parietal alpha sub Quiet (t = 0.629 p = 0.535) variables.

Concerning the potential role of the onset of deafness on the investigated EEG indices, given the effect of the variable Group on F8 alpha SNR sub Quiet and Parietal alpha SNR sub Quiet, a comparison was performed between a group composed by pre- and peri-lingual deaf CI users and the group composed by post-lingual deaf CI users. Results did not report any statistically significant difference for F8 alpha SNR sub Quiet (t = 0.469 p = 0.643) neither for Parietal alpha SNR sub Quiet (t = 0.710 p = 0.484).

Finally, no correlation between period of CI use and alpha activity in F7, F8 and parietal area was found, and neither in the SNR "sub Quiet" condition (obtained for each of the three areas by the subtraction from the alpha activity corresponding to the listening to the SNR5 and SNR10 conditions of the alpha activity corresponding to the Quiet condition).

## Discussion

In general, results showed that CI users performed worse than NH controls in the recognition of the emotional content of the stimuli (Fig 1), while reaction times did not appear to be influenced by the factor group (Fig 2). Furthermore, when considering the alpha activity corresponding to the listening to signal to SNR5 sub Quiet and SNR10 sub Quiet conditions— ideally removing the emotional content of music and isolating the difficulty level due to the SNRs- CI users reported higher levels of activity in the parietal alpha and in the homologous of the left IFG in the right hemisphere (F8 EEG channel), in comparison to NH (Fig 5).

Moreover, the percentage of correct responses, the alpha F8 SNR sub Quiet and the period of CI use all presented a similar pattern of correlation with the percentage of musical excerpts rated as of Easy and Medium listening difficulty (Fig 6).

Reaction times, decreasing over trials, appeared to be more related to the novelty of the musical stimuli [67], as suggested by the order effect reported in such analysis, in fact reaction times are the most common behavioural measure of speed of processing [68]. Therefore, such results support the evidence that more novel stimuli are processed more slowly than already met ones, and presumably with less effort as indexed by the speed of processing. Differently, the percentage of correct responses was more related to the factor group, given the statistical significant difference between NH and UCI for such index, and specifically with higher percentage of correct responses in the emotional categorization of the musical stimuli for the NH group in comparison to the UCI group, as expected given the lower performances.

The direct correlation between percentage of Correct Responses and percentage of musical pieces rated as Easy to listen to and the inverse correlation between percentage of Correct Responses and percentage of musical pieces rated as of Medium difficulty to listen to only in UCI could reflect a certain degree of consciousness about the effort applied by patients in attending the auditory task, witnessed also by other studies concerning listening effort [69]. However the lack of correlation between percentage of Correct Responses and Difficult to listen to musical pieces could reflect an underestimation of the level of the effort needed to perform the task, often observable in self-declared ratings [24], in fact the UCI group reported lower performances than the NH group. Indeed, such evidence is in accordance with the existence of an imperfect relationship between subjective measures of effort and cognitive measures [70], and these latter are strictly linked to performances [71].

The difference in the perceived pleasantness ratings with the higher percentage of musical excerpts rated as Liked by NH group in comparison to the UCI group is easy explained by the increased enjoyed musical fruition by persons with a physiologically functioning auditory system in comparison to CI users, due to unavoidable technical constriction of the CI device [17, 72].

The lack of statistical significances concerning the influence of the side of the CI on behavioural and neurophysiological correlates are in accordance to previous evidences of the side of the implantation as being irrelevant with respect to emotional categorization tasks in CI user children [63], and support the hypothesis of a major influence of the anatomical functioning specialization for subserving such processing [59]. The lack of differences within the UCI group on the base of the onset of the deafness in relation to F8 alpha SNR sub Quiet and Parietal alpha SNR sub Quiet indices, supports the hypothesis of a lack of influence of such variables on such indices, and therefore on the concurrent cognitive task execution. However, due to the poor numerosity of the two subgroups (pre- and peri-lingual deaf CI users n = 12; post-lingual deaf CI users n = 15), the result would need further investigations, also eventually expanding the included EEG indices.

The correlations between alpha activity sub Quiet and rated effort observed only in F8 area could be explained by a specialization of this area for musical stimuli, resembling the observed correlation between alpha activity in parietal region and rated effort, interestingly following a quadratic relation where alpha power values were decreased when effort ratings were at the top and bottom of the effort rating scale, and increased when effort ratings were in the centre of the scale [73]. The CI use has been previously suggested to influence neuroplasticity both in children [50, 63, 74, 75] and adults [76–78], especially in auditory and visual areas. sub Quiet Results did not evidence a correlation between the period of CI use and alpha activity, however the correlation between the period of CI use and difficulty ratings (negative correlation with musical pieces rated as Easy to listen to and positive correlation with musical pieces rated as of

Medium difficulty to listen to) exactly reflects the correlation between F8 sub Quiet alpha and difficulty ratings (Fig 6). This could be an indirect suggestion of a modulation due to the period of CI use on the brain activity linked to listening effort. Accordingly, an influence of the period of CI use in adult CI users was previously shown in an event related potential study, and specifically on the N100 amplitude and on the P300 latency [79]. Moreover, a more recent CI activation could be linked to a more sustained employment of residual low-frequency hearing [16], in particular in the non-implanted ear (that is the most of the times the ear with more residual hearing [80]), in comparison to longer period CI users, possibly resulting in a more easy fruition of music.

Happy musical pieces, irrespective of the group, produced higher levels of alpha activity in comparison to sad excerpts, in all the investigated areas (parietal, F7 and F8) and in parallel reported a higher percentage of Correct Responses in both groups. This latter result is in accordance with previous studies conducted on the same musical database reporting a more effective recognition of happy emotional tone in musical stimuli in comparison to sad ones [81]. Probably, in order to perform the categorization task better, more listening effort would be needed, as indexed by alpha levels [42, 46, 49, 56]. In addition, the fact that happy stimuli induced higher alpha activity in comparison to sad ones is in accordance with previous evidence comparing happy conditions to negative emotional conditions [82]. The study just mentioned was conducted on healthy persons, therefore the present study, does not provide evidence of a group effect and supports the hypothesis that such phenomenon is preserved in UCI adults, despite the showed decrease in emotion recognition performances in comparison to NH controls. In contrast, EEG-based evidences of different processes in auditory emotion recognition have been provided in UCI children [63], as well affected by auditory emotion recognition deficits in comparison to NH controls.

The effect of the emotional content of the musical pieces observed in F7 but disappearing when subtracting from the activity corresponding to the SNR5 and SNR10 the Quiet condition, that ideally represents the emotional content of the musical excerpts, is aligned with the notion that such area is implied in the processing of the semantical features of music [10]. Whilst the fact that activity in F8, when subtracted of the Quiet condition, showed higher activity for UCI in comparison to NH, as well as parietal area, suggests that both these areas are implied in listening effort related processes with musical stimuli. The parietal area, in fact is well-known to be involved in listening effort with speech and tone stimuli [25, 46, 49]. However, a novel result concerns the role of alpha activity localized in F8, appearing to be particularly sensitive to listening effort processes related to musical stimuli. In fact, an effect of the audibility condition in such area was reported here, and specifically lowest alpha activity and therefore listening effort levels were here produced by the listening to the SNR5 condition, that is the most difficult condition included into the present study. This could be explained by a renounce phenomenon, due to excessive demand from the task, leading to a lack of increase in alpha activity and listening effort, already reported in some studies [43, 44, 46]. Alternatively, some studies suggested that alpha was more suppressed during more demanding listening conditions [39, 48, 83]. This is as well in accordance with present results, since the most demanding condition (SNR5) produced lower alpha levels. Moreover it is interesting to note that F8 has been showed to be involved in musicogenic epilepsy (epileptic seizures triggered by music) [84], further supporting the hypothesis of a peculiar role for such area in musical features processing.

The fact that, in contrast to speech in noise recognition studies showing the involvement of F7 in listening effort-related phenomena [56], in the present study, after the subtraction of the alpha activity recorded during the quiet condition from the one recorded during the conditions characterized by background noise, could be explained by the classical notion of the

major specialization of the right hemisphere for music in comparison to the left one [85]. But beyond that, a further explanation could be in the evidence that despite a common involvement of both F7 and F8 in the syntactic processing of music, the right counterpart (F8) presented a stronger response during such activity, suggesting a greater specialization of F8 for music processing and of F7 for language [57]. In fact, the already evidenced involvement of F7 in listening effort was tested employing speech stimuli [56], and the existence was suggested of differential executive networks subserving listening effort [86]. In line with this perspective of specialized regions for different aspects of listening effort, the present results support the hypothesis of a major specialization of F8 for listening effort in response to musical stimuli, showing a sensitivity among SNRs after the subtraction of the alpha activity corresponding to the listening in quiet. Accordingly, previous fNIRS studies, focusing on the influence on the lateralization of music processing, employed protocols presenting music only and music mixed with noise stimuli, showed an increased right lateralization in the condition with noise [34].

In conclusion, the present study supports the hypothesis of the application of listening effort studies also to music, with possible implications also in rehabilitation, given the employment of music in such kinds of interventions [13, 14, 87]. Interestingly there was no difference between NH and UCI in the listening effort when considering the influence of the emotional tone of music, but the processing of the musical stimuli excluding emotional influences are affected by background noise disturbing activity in UCI participants, with a possible modulation exerted by the time of CI use, but not by the side of the implantation. In particular for listening effort study purposes in music, the right counterpart of Broca's area appears more sensitive in comparison to the left Broca's and parietal areas, more traditionally linked to speech-related listening effort.

## Methods

### Participants

For the study 27 UCI patients (14 F, 13 M, mean age 46.185 ± 12.263) were recruited, 16 implanted in their left ear and 11 in their right ear, at the moment of the test, none of them wore any hearing aid in their contralateral ear. All patients were right-handed except one. All patients underwent cochlear implant surgery in adulthood and were characterized by a different onset of deafness, despite a typical predominance of postlingual deafness etiology in adults [88]: 8 prelingual, 4 perilingual (defined as an onset of deafness at 2–3 years of age), 15 postlingual deafness acquisition. The audiometry-related inclusion criterium for UCI was word comprehension rate of at least 50% at 65 dB SPL [19] and, that intensity was used for stimuli delivery in the experiment. The 50% threshold has been set because a common measure of the ability of a listener to understand speech in noise is the speech reception threshold (SRT) [89], defined as the SNR where 50% of the speech is correctly understood. In addition, 10 NH (average threshold for pure tone frequencies 250–4000 Hz ≤ 20 dB HL) controls (5 F, 5M, mean age 43.300 ± 14.514), all right-handed, were enrolled into the study.

### Stimuli

Stimuli were excerpts of classical piano music, previously categorized as happy (e.g. Beethoven's Symphony No. 6) or sad (e.g. Albinoni's Adagio) in a database made available by authors (63) and already employed in studies concerning the recognition of musical features by CI users [17, 81]. From the original database of 32 musical excerpts, after a pre-test conducted on NH university students (n = 32) concerning the recognizability of the emotional content of the excerpts, in order to exclude the ones characterized by the poorest recognition 2 lists of 24

items have been created, with an average duration of 15.386 ±5.617 seconds per excerpt. Each list included 24 musical stimuli, composed by 8 musical excerpts, half happy and half sad, belonging to the original database delivered in Quiet (Q), and the same excerpts delivered with background noise, continuous 4-talker babble background noise at SNR10 and SNR5, being the SNR5 the most difficult audibility condition among the presented ones. Stimuli were delivered free-field through two loudspeakers placed in front of and behind the participant at a distance of 1 meter each [63], so to meet CIs best requirements for their use, at an average intensity of 65 dB SPL, measured at the subject's head [17, 44]. None of the participants was a musician or was affected by psychiatric or neural diseases, nor used drugs with psychoactive effect at least in the six months preceding the experiment. UCI participants were also asked about their habits concerning mean hours of listening to music per week [72], resulting in the majority of the sample (55.56%) as reserving 0–2 hours per week to such activity; the least percentage of responders (both 7.407%) reserved 3–4 hours/week and >9 hours/week; 11.111% of the responders answered 5–6 hours/week for; finally, 18.518% of UCI participants preferred to not respond to such question.

Stimuli were delivered in a randomized order through E-prime software (Psychology Software Tools, Inc., USA), that allowed also the collection of behavioural data.

## Protocol

Participants underwent a familiarization with the protocol, employing musical stimuli not employed in the real study, but belonging to the same database of the experimental ones. Participants, equipped with the EEG cap, were sitting in front of a computer, instructed to listen to musical stimuli and to limit movements as much as possible. Each musical stimulus (average duration ± st.d.: 15385 ± 5617 ms) was preceded by a white screen (1500 ms) and a grey screen with a fixation cross (3000 ms). After each musical excerpt on the screen simultaneously happy and sad face drawings appeared already used in other studies conducted on the same musical database [81, 90], and through the use of a customized keyboard participants were instructed to press the button below each face in order to assign the presumed emotional content to each excerpt. Half of the correct responses corresponded to the right button and half to the left button. After that, in order to make participants rate their personal difficulty at the end of each listening, on the screen appeared the drawing of a sort of speedometer with three icons below characterized by different evocative colours: "Easy" (green), "Medium" (yellow) and "Difficult" (red). After that, for the rating of the perceived pleasantness of each musical piece, appeared three icons that the participant had to select: thumbs up (I like), thumbs horizontal (Indifferent) and thumbs down (I do not like). These three point scale were chosen for their convenience, especially in impaired participants [91, 92]. Participants were requested to employ the touchpad in order to press the selected icon. There was no time limit for giving any responses. Participants were instructed to use their favourite hand for giving responses either through the keyboard buttons and the touchpad and after giving each response to return to the same rest position, in order to homogenise the starting position of each trial and subtrial phase.

The study was carefully explained to all participants, specifying that they would not receive any form of compensation for their participation and that they were able to leave the experiment at any moment with no need of giving explanation for such decision. Participants, after being allowed to make all questions they had to the experimenter, signed a written informed consent to the participation. The study was approved by the Gemelli Hospital Ethical Committee, and was conducted according to the principles outlined in the Declaration of Helsinki of 1975, as revised in 2000.

## Behavioural data

Concerning behavioural data, included into the study were: percentage of Correct Responses, based on the recognition of the emotional tone of each musical piece; Reaction Times; rating concerning the difficulty (three levels [93, 94]: Easy, Medium, Difficult) perceived by participants in listening to each musical excerpt. Additionally, concerning demographic data, it was included into the analysis the period of CI use (in months) at the time of the experiment performance.

## EEG data

A digital EEG system (Beplus EBNeuro, Italy) was used to record 20 EEG channels (Fpz,Fz,F3, F4,F7,F8,Cz,C3,C4,Cp5,Cp6,T7,T8,Pz,P3,P4,P7,P8,O1,O2) according to the international 10/ 20 system, with a sampling frequency of 256 Hz. The impedances were maintained below 10 kΩ, and a 50 Hz notch filter was applied to remove the power interference. A ground electrode was placed on the forehead and reference electrodes on earlobes. The EEG signal was initially band-pass filtered with a 5th-order Butterworth filter (high-pass filter: cut-off frequency fc = 1 Hz; low-pass filter: cut-off frequency fc = 40 Hz). Through the application of a regression-based method, eyeblinks artifacts have been identified and corrected. In particular, the Fpz channel was used to identify and remove eye-blink artifacts by the use of the REBLINCA algorithm [95]. For other sources of artifacts (e.g. environmental noise, user movements, etc.) specific procedures of the EEGLAB toolbox were employed [96]. In particular, the EEG dataset was firstly segmented into epochs of 2 s through moving windows shifted by 0.125 s. This windowing was chosen with the compromise of having both a high number of observations, in comparison with the number of variables, and in order to respect the condition of stationarity of the EEG signal. This is in fact a necessary assumption in order to proceed with the spectral analysis of the signal. Then, three criteria were applied to those EEG epochs [97, 98]: i)Threshold criterion (amplitudes exceeding ±100 µV); ii) Trend criterion (slope higher than 10 µV/s); iii) Sample-to-sample criterion (sample-to-sample amplitude difference higher than 25 µV).

All EEG epochs marked as "artifact" were removed in order to have a clean EEG signal. In order to accurately define EEG bands of interest, for each participant the Individual Alpha Frequency (IAF) was computed on a closed eyes segment recorded prior to the experimental task. Thus, the EEG was filtered in the alpha [IAF-2 ÷ IAF+2 Hz] [99]. EEG recordings were segmented into trials, corresponding to the listening of each musical excerpt (ranging from approximately 8000 ms to 28000 ms), and excluding all experimental phases, beyond the listening to music, potentially affected by muscular artifacts or cognitive processes for instance linked to the difficulty rating phase. The Power Spectrum Density was calculated in correspondence of the different conditions with a frequency resolution of 0.5 Hz. Trials were normalized by subtracting the open eyes activity recorded before the beginning of the experimental task.

For the calculation of the EEG alpha activity over the parietal area included in the analysis was the estimated PSD deriving from the following channels: Pz,P3,P4,P7,P8. For the analysis of the EEG alpha activity in the left inferior frontal gyrus (including the Broca's area) and its right homologous were included respectively the channel F7 and F8 [54].

## Statistical methods

Sample size calculation was performed in order to a priori select the minimum number of participants to be enrolled, resulting to be 28. In order to perform the sample size analysis, G*Power 3.1 [100] was used, a free software developed by HHU—Düsseldorf University, selecting as type of power analysis "A priori: Compute required sample size given α, power and effect size, and setting significance level α = 0.05, power (1-β) = 0.95 and the by default

medium effect size f = 0.25 [101, 102]. In the present study 27 UCI and 10 NH participants were enrolled, and in order to verify the requested assumption of homogeneity between the variances of the two groups for ANOVA calculation Levene's F test was performed, that resulted not significant, therefore verifying the assumption. In one case Levene's test resulted significant, therefore Welch's test was performed in order to verify the suitability of the ANOVA test for present data collection, that resulted not significant.

The UCI group was divided into two groups: pre- and peri-lingual deaf CI users and post-lingual deaf CI users. This solution was applied in order to compare these two groups for alpha F8 SNR sub Quiet and for Parietal alpha SNR sub Quiet indices.

In the analysis concerning the rated difficulty about the listening to each musical piece, the UCI group was reduced to 18 participants, due to a malfunction causing lack of data recording by the software employed for the delivery of stimuli and the recording of responses. In addition, the number of CI users was reduced in comparison to the full sample in the analyses concerning the music listening habits, because of the just mentioned malfunction and to the expressed preference by some participants of not responding to such a question.

Repeated measure ANOVA were performed in the comparisons between groups for the different EEG indices (F7 alpha, F8 alpha and parietal alpha, and the same three indices with SNR5 and SNR10 conditions subtracted of the Quiet condition) and for behavioural data (Correct Responses, Reaction Time). Three factors were investigated: Group (two levels: UCI, NH), Emotion (two levels: Happy, Sad) and SNR (three levels: Quiet, SNR5, SNR10; or alternatively two levels: SNR5 sub Quiet; SNR10 sub Quiet). In the first part of EEG results it has been showed that for all F7, F8 and Parietal area alpha activity subtracted of the Quiet condition there was no effect of the different SNR condition (SNR5, SNR10—as reported in the text), but there was an effect of the factor Group only after the subtraction of the Quiet condition from the two separate SNRs conditions both for F7 and Parietal area. This suggested authors that the presence of background noise, irrespective of the two particular SNRs conditions chosen for the present study, could highlight relevant differences between the two groups. This led authors to the choice of investigating the average of the two SNR condition subtracted of the Quiet condition for further ANOVA analyses and correlation analyses with behavioural data. Repeated measure ANOVA test was also employed for the investigation of an order effect in correspondence of the songs repetition under the three different SNR conditions; in particular three factors were included into the analysis: Group (two levels: UCI, NH), Emotion (two levels: Happy, Sad) and Order (three levels: first, second, third). Moreover, through repeated measure ANOVA were analysed: rated difficulty (% or excerpts rated as Easy, Medium, Difficult), rated appreciation (% or excerpts rated as Liked, Indifferent, Disliked), reaction times and percentage of Correct Responses, considering into the analysis the factors Group, Emotion and SNR or Order (first, second and third listening to the same music).

Correlations were investigated through Pearson's test between and among EEG indices and behavioural data [103–105].

## Supporting information

**S1 Annex.**
(DOCX)

**S2 Annex.**
(DOCX)

**S1 File.**
(PDF)

## Author Contributions

**Conceptualization:** Giulia Cartocci, Fabio Babiloni.

**Data curation:** Giulia Cartocci, Bianca Maria Serena Inguscio, Andrea Giorgi, Rosa Grassia, Tiziana Di Cesare, Francesco Freni, Roberto Albera, Andrea Canale.

**Formal analysis:** Giulia Cartocci, Bianca Maria Serena Inguscio.

**Funding acquisition:** Fabio Babiloni.

**Investigation:** Bianca Maria Serena Inguscio, Andrea Giorgi, Roberto Albera, Andrea Canale.

**Methodology:** Alessia Vozzi, Rosa Grassia.

**Resources:** Tiziana Di Cesare.

**Software:** Alessia Vozzi.

**Supervision:** Carlo Antonio Leone, Walter Di Nardo, Anna Rita Fetoni, Francesco Freni, Francesco Ciodaro, Francesco Galletti, Roberto Albera, Andrea Canale, Lucia Oriella Piccioni, Fabio Babiloni.

**Writing – original draft:** Giulia Cartocci.

**Writing – review & editing:** Bianca Maria Serena Inguscio, Carlo Antonio Leone, Walter Di Nardo, Anna Rita Fetoni, Francesco Ciodaro, Francesco Galletti, Roberto Albera, Andrea Canale, Lucia Oriella Piccioni, Fabio Babiloni.

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
