## [Decision Letter · Decision Letter 0]

26 Dec 2022

PONE-D-22-15819Music in noise recognition: an EEG study of listening effort in cochlear implant users and normal hearing controlsPLOS ONE

Dear Dr. Cartocci,

Thank you for submitting your manuscript to PLOS ONE. After careful consideration, we feel that it has merit but does not fully meet PLOS ONE’s publication criteria as it currently stands. Therefore, we invite you to submit a revised version of the manuscript that addresses the points raised during the review process. Please submit your revised manuscript by Feb 09 2023 11:59PM. If you will need more time than this to complete your revisions, please reply to this message or contact the journal office at plosone@plos.org. Please include the following items when submitting your revised manuscript:A rebuttal letter that responds to each point raised by the academic editor and reviewer(s). You should upload this letter as a separate file labeled 'Response to Reviewers'.A marked-up copy of your manuscript that highlights changes made to the original version. You should upload this as a separate file labeled 'Revised Manuscript with Track Changes'.An unmarked version of your revised paper without tracked changes. You should upload this as a separate file labeled 'Manuscript'.

We look forward to receiving your revised manuscript.

Kind regards,

Paul Hinckley Delano, Ph.D.

Academic Editor

PLOS ONE

https://journals.plos.org/plosone/s/fileid=ba62/PLOSOne_formatting_sample_title_authors_affiliations.pdf.2.

Reviewers' comments:

Reviewer's Responses to Questions

**Comments to the Author**

1. Is the manuscript technically sound, and do the data support the conclusions?

Reviewer #1: Yes

Reviewer #2: Yes

2. Has the statistical analysis been performed appropriately and rigorously? 

Reviewer #1: Yes

Reviewer #2: Yes

3. Have the authors made all data underlying the findings in their manuscript fully available?

Reviewer #1: Yes

Reviewer #2: No

4. Is the manuscript presented in an intelligible fashion and written in standard English?

Reviewer #1: Yes

Reviewer #2: No

5. Review Comments to the Author

Reviewer #1: The present manuscript is adresses a research topic of great interest, nevertheless some important issues in how it's findings are presented should be solved before considering publication:

1) There are some errors in the results section. The first and most important is thar in the very first sentence NH groups is compared with NH group. One should be UCI group.

2) Please describe abbreviation meanings when first appearing in the text, (even if it has been clarified in the abstract section). This occurs with SNR and UCI (clarify the complete meaning of Unilateral cochlear implant group).

3) There are important issues with the graphical representation of the results. I suggest using boxplots more than bars with error margins: Moreover,

4) Error margins, particularly in figure 1, extend over 100%. This is confusing, I understand is a producto of the standard deviation analysis, but given a maximal of 100% in that metric, It is preferable to use box-plot.

5) Please do not use Abbreviation when avoidable in the title of a Figura: CR = Correct Responses.

6) Please avoid the use of "%", use Percentage of...

7) I missed seing a figure of Response Time, segregated between NH vs UCI groups, and subdiveded in the SNR5 SNR10 scenarios. The authors mention a lack of significance in this differences, but considering the article is based upon MUSIC in NOISE recognition, the focus of results should be in that... Recognition in diferent Noise settings. Also compared with no-noise setting.

8) In figure 3, it is not proper to use a Line graph between the two group of analysis, which implies a change of condition, for example temporal or pre-post something in a same group. Use boxplots.

9) In figure 4 the grouping of %Easy vs %Medium seems inapropiated, without the %Difficult, and it seems that % Medium is 1-%Easy whitch would make the grouping irrelevant, and the complete opposite correlation redundant (also the supperimposed mirror-like graph is confusing. Consider leaving only %East group, and improve the overall quality of each different sub-figure, also adding in the figure a value of the correlation coefficients.

Reviewer #2: General comments:

The authors present interesting findings and generally have a sound experimental design. However, their ability to communicate these findings is impeded by many grammatical errors. Significant editing is needed to improve the flow and readability of the manuscript’s language.

The aims of the study are communicated, but the authors do not make clear predictions regarding what they are expecting to find in alpha results for F7/F8 or parietal regions for UCI vs. NH participants, effects of CI side, etc. If the authors did have preliminary hypotheses regarding these findings, it would be ideal to address these in the introduction and tie them to their discussion of previous findings regarding the regions involved in listening effort and emotional processing. If not, an explicit statement that the study was exploratory is needed.

Referring to F8 as Broca’s area and F7 as the right hemisphere homologue of F8 is misleading, as the spatial resolution of a standard EEG system is not fine enough to identify Broca’s area specifically. The Koessler et al. (2009) paper that is cited to justify choosing F8 and F7 for analysis labels these electrodes as left and right inferior frontal gyrus. I recommend using “left/right inferior frontal gyrus” instead of the current terminology surrounding Broca’s area, as it is a more accurate representation of the regions underlying those electrode sites.

The current results section reads a bit like a laundry list of statistics. It would be helpful to describe the patterns found in the data for each result before presenting and interpreting the relevant statistic. Relatedly, there are several significant behavioral results discussed that do not have an accompanying figure or table to indicate the direction of the effect. Any significant result that is discussed in the manuscript should have a visualization of that result. Results that are not significant do not need a figure, nor do they need the non-significant p-value reported.

The CI sample for the study seems to have some big differences in hearing history and CI use (e.g., prelingual vs. postlingual deafness, word comprehension abilities). I would be interested to see more analyses that test whether these demographic characteristics beyond CI side and duration of use are related to differences seen in alpha in F7/F8.

The current figures could be improved to be more publication-ready. For example, the caption for Figure 2 references asterisks to indicate significant differences, but they do not appear in the figure. Consider using an alternative to bar graphs that could better indicate the variability in the data (e.g., violin plots), especially given that CI recipients tend to be a very heterogeneous group.

Some unnecessary abbreviations are used for common phrases, which I found unnecessary. For example, using “CR” instead of saying “correct responses.” Using abbreviations only where it improves the flow or in situations where that abbreviation is common (e.g., SNR, UCI, NH) will help with readability.

The manuscript needs to be checked for adherence to the journal’s style requirements. For example, in the reference list, I noticed that there are months included in the article references that are not written in English (the date of publication is not actually required for ICJME style for journal articles, the year is sufficient).

Specific comments:

Line 119—“NH group performed better recognition of the emotional content of the musical pieces in comparison to NH group.” –NH group is repeated twice.

Line 146—A figure or table should be used to represent the results of difficulty rating.

Line 160—Correlations are reported, but not their significance level. I’d recommend running a correlation test for each of these relationships, and then reporting the p values for the relationships that are significant. The correlation test accounts for not only the strength of the correlation, but also the sample size. (In R, this is done using cor.test() ).

Line 167—A figure or table should be used to represent the results of perceived pleasantness.

Line 189—I am unsure about the choice to subtract the alpha activity for the Quiet condition from an average of the SNR5 and SNR10 conditions. Please provide a justification here or in the method section for why this was done, as opposed to comparing SNR5 – Quiet and SNR10 – Quiet separately.

Line 210—As I suggested for the behavioral results, please include results of the correlation test with significant p values reported. I suspect that some of these correlations may not be significant—especially the correlation in the third panel of Figure 4, given that there are few participants with long-term CI use.

Line 237—For the discussion section, it would be helpful to first summarize the major behavioral and neural findings before moving to discussion of specific findings, rather than jumping into talking about reaction time right away.

Lines 268-272—“Despite the lack of statistically significant correlation between period of CI use and alpha activity, the negative correlations between the period of CI use and the percentage of musical pieces rated as Easy to listen and vice versa the positive correlation with the percentage of musical pieces rated as of Medium difficulty to listen, exactly reflects results obtained through the assessment of the correlation between F8 sub Q and difficulty ratings.” This is a very long sentence, and the point the authors are trying to make is not clear about how these correlations are related.

Lines 336-347—With a highly variable group like CI recipients, it would be helpful to include a table to indicate individual characteristics of hearing history and CI use for each person (this could be included as a supplement if desired).

Lines 362-366—It is reported that participants were asked about their music listening habits, but those habits were not considered in the analysis of behavioral or neural data. Consider adding in some analysis of this variable (does frequency of music listening correlate with accuracy of emotion identification, for example?), or remove discussion of the question from the methods.

Line 374—What was the duration (or average and standard deviation of the duration) of the musical stimuli?

Line 420—“EEG recordings were segmented into trials, corresponding to the listening of each musical excerpt.” –This statement is unclear. How long was each of the trial epochs?

Line 248—Provide more explanation about how the sample size calculation was performed.

6. PLOS authors have the option to publish the peer review history of their article (what does this mean?). If published, this will include your full peer review and any attached files.

Reviewer #1: **Yes: **Hayo A. Breinbauer

Reviewer #2: No

---

## [Author Response · Author response to Decision Letter 0]

17 Feb 2023

Reviewer #1: The present manuscript is adresses a research topic of great interest, nevertheless some important issues in how it's findings are presented should be solved before considering publication:

1) There are some errors in the results section. The first and most important is thar in the very first sentence NH groups is compared with NH group. One should be UCI group.

We apologise for the typo and thank the Reviewer for highlighting it. We corrected it into the revised manuscript:

“NH group performed better recognition of the emotional content of the musical pieces in comparison to UCI”

2) Please describe abbreviation meanings when first appearing in the text, (even if it has been clarified in the abstract section). This occurs with SNR and UCI (clarify the complete meaning of Unilateral cochlear implant group).

We thank the Reviewer for the suggestion and modified the very first sentence where “unilateral cochlear implant” appears, and explained the meaning of UCI as follows: 

“Such deficits have been showed to be reflected also by different cerebral activity patterns in comparison to NH controls, that were also reported to be influenced by deafness etiology and by unilateral or bilateral CI (UCI and BCI respectively) conditions, that is the condition of having and using the CI in only one or in both ear sides”

Also for “SNR” we introduced the acronym in the main text according to the Reviewer’s suggestion, and we thank him for highlighting it.

3) There are important issues with the graphical representation of the results. I suggest using boxplots more than bars with error margins: Moreover,

4) Error margins, particularly in figure 1, extend over 100%. This is confusing, I understand is a producto of the standard deviation analysis, but given a maximal of 100% in that metric, It is preferable to use box-plot.

We thank the Reviewer for the suggestion and we modified accordingly, please see the New Figure 1 below, also included into the revised manuscript.

5) Please do not use Abbreviation when avoidable in the title of a Figura: CR = Correct Responses.

6) Please avoid the use of "%", use Percentage of...

Point 5 and 6: We thank the Reviewer for the suggestion and we modified accordingly, please see the New Figure 1 below, also included into the revised manuscript.

New Figure 1. Left: box plot reporting the percentage of Correct Responses in the comparison between normal hearing (NH) and unilateral cochlear implant users (UCI). Right: box plot reporting the percentage of Correct Responses in the comparison between Happy and Sad musical pieces. Whiskers from minimum to maximum value; 25th, 50th (median) and 75th percentile drawn as horizontal lines in each box; “x” stands for the mean.

7) I missed seing a figure of Response Time, segregated between NH vs UCI groups, and subdiveded in the SNR5 SNR10 scenarios. The authors mention a lack of significance in this differences, but considering the article is based upon MUSIC in NOISE recognition, the focus of results should be in that... Recognition in diferent Noise settings. Also compared with no-noise setting.

We thank the Reviewer for the comment, and accordingly we modified the Figure 2 (please see the New Figure 2 below), reporting on the left the reaction times reported by each group (UCI and NH) in each experimental auditory condition (Quiet, SNR5 and SNR10). Moreover, the fact that concerning reaction times we did not find any difference between the groups, whilst it was found on the base of the order of presentation of the musical stimuli, could be explained by the higher effect of the repetition, and therefore of an easier and faster reaction to ever more “familiar” stimuli, as also suggested by the cognitive fluency theory (e.g. Huron, D. (2013). A psychological approach to musical form: The habituation–fluency theory of repetition).

Figure 2. Left: Representation of the mean Reaction Time for the emotional recognition task execution of the musical excerpts, focusing on the different auditory conditions (Quiet, SNR10, SNR5). Right: Representation of the mean Reaction Time for the emotional recognition task execution of the musical excerpts, focusing on their order of appearance and irrespectively of the audibility condition (Q, SNR5, SNR10). Whiskers from minimum to maximum value; 25th, 50th (median) and 75th percentile drawn as horizontal lines in each box; “x” stands for the mean

8) In figure 3, it is not proper to use a Line graph between the two group of analysis, which implies a change of condition, for example temporal or pre-post something in a same group. Use boxplots.

We thank the Reviewer for the suggestion, we modified accordingly, please see the new figure 5 (ex Figure 3) below:

New Figure 5. Left up: comparison between the UCI and NH groups mean values of alpha F8 activity calculated subtracting the activity recorded during the listening to the Quiet condition from the activity corresponding to the listening to the conditions characterized by the presence of background noise. Whiskers from minimum to maximum value; 25th, 50th (median) and 75th percentile drawn as horizontal lines in each box; “x” stands for the mean. Left down: comparison between the UCI and NH groups mean values of alpha parietal activity calculated subtracting the activity recorded during the listening to the Quiet condition from the activity corresponding to the listening to the conditions characterized by the presence of background noise. Whiskers from minimum to maximum value; 25th, 50th (median) and 75th percentile drawn as horizontal lines in each box; “x” stands for the mean. Right: topoplot representing the t values calculated comparing the mean gamma PSD values for each channel included into the statistical analysis reported into the graphs on the left (F8 and Pz, P3, P4, P7, P8 respectively). 

9) In figure 4 the grouping of %Easy vs %Medium seems inapropiated, without the %Difficult, and it seems that % Medium is 1-%Easy whitch would make the grouping irrelevant, and the complete opposite correlation redundant (also the supperimposed mirror-like graph is confusing. Consider leaving only %East group, and improve the overall quality of each different sub-figure, also adding in the figure a value of the correlation coefficients.

We thank the Reviewer for the comment and for the opportunity to better explain our results, we also apologise for the lack of clarity. At first in the Fig.4 we decided to report “%Easy” and “%Medium” because of the opposite patterns showed by them, but both reporting statistically significant correlations, differently from “%Difficult” that due to the lack of statistical significance was not graphicated. However we understand the point of the Reviewer about the major informativity conferred by reporting in the graph also “%Difficult” and values of the correlations coefficients, so we added those items in the new Figure 4 (below). We also just wanted to ensure the Reviewer about the calculation of the percentages of musical excerpts rated as Easy, Medium and Difficult to listen: %Medium was not calculated as 1-%Easy. In fact, for each participant to the study, all percentages were calculated starting from the number of musical excerpts rated as Easy, Medium or Difficult over the total number of musical excerpts (n=24). For instance: if one participant rated 8 musical excerpts as Difficult to listen, it would correspond to (8/24)x100=33.33%.

New Figure 6 (ex Figure 4)

Reviewer #2: General comments:

The authors present interesting findings and generally have a sound experimental design. However, their ability to communicate these findings is impeded by many grammatical errors. Significant editing is needed to improve the flow and readability of the manuscript’s language.

We thank the Reviewer for the comment and apologise for the grammatical errors, we had the revised version edited by a native English speaker.

The aims of the study are communicated, but the authors do not make clear predictions regarding what they are expecting to find in alpha results for F7/F8 or parietal regions for UCI vs. NH participants, effects of CI side, etc. If the authors did have preliminary hypotheses regarding these findings, it would be ideal to address these in the introduction and tie them to their discussion of previous findings regarding the regions involved in listening effort and emotional processing. If not, an explicit statement that the study was exploratory is needed.

We thank the Reviewer for the comment and we apologise for the lack of clarity. In fact, actually we presumed to obtain some differences between CI users and NH controls, given some literature references reported about IFG and Parietal alpha. In order to make more explicit such hypothesis we added the following sentences in the introduction section:

The sum of the just mentioned evidences reported above and concerning IFG and Parietal alpha activities as indices of listening effort, supported their sensitivity in highlighting differences between CI users and NH controls. This led to the present hypothesis of obtaining differences between the two groups also in the current study, involving musical stimuli with background noise.

The left IFG includes canonical language production regions such as Broca’s area, known to be activated during covert or internalized speech production40 and working memory involving speech41.

Referring to F8 as Broca’s area and F7 as the right hemisphere homologue of F8 is misleading, as the spatial resolution of a standard EEG system is not fine enough to identify Broca’s area specifically. The Koessler et al. (2009) paper that is cited to justify choosing F8 and F7 for analysis labels these electrodes as left and right inferior frontal gyrus. I recommend using “left/right inferior frontal gyrus” instead of the current terminology surrounding Broca’s area, as it is a more accurate representation of the regions underlying those electrode sites.

We thank the Reviewer for the comment. We understand that considering F7 as Broca’s area and F8 its contralateral counterpart represents a simplification, however, beyond the article by Koessler et al. (2009), we employed that correspondence because of the same simplification previously reported by Dimitrijevic and colleagues (Dimitrijevic, A., Smith, M. L., Kadis, D. S., & Moore, D. R. (2019). Neural indices of listening effort in noisy environments. Scientific Reports, 9(1), 1-10). These authors justified that by the sentence: “The left IFG includes canonical language production regions such as Broca’s area, known to be activated during covert or internalized speech production and working memory involving speech”. However, according to the Reviewer’s comment, we replaced “Broca’s area” with “left/right inferior frontal gyrus (IFG)”, as suggested by the Reviewer, and we modified in the abstract the sentences were Broca’s area was mentioned:

“Listening effort was investigated, in addition to participants’ rating and performance, employing EEG features known to be involved in such phenomenon, that is alpha activity in parietal areas and in the left inferior frontal gyrus (IFG), that includes the Broca’s area.” 

 “Furthermore, when considering the alpha activity corresponding to the listening to signal to noise ratio (SNR) 5 and SNR10 conditions subtracted of the activity while listening to the Quiet condition - ideally removing the emotional content of music and isolating the difficulty level due to the SNRs- CI users reported higher levels of activity in the parietal alpha and in the homologous of the left IFG in the right hemisphere (F8 EEG channel), in comparison to NH.”

Accordingly, in the introduction we changed:

First aim of the present research was the investigation in NH and unilateral UCI participants of the occurrence of listening effort-related cerebral patterns, indexed by the modulation of alpha activity in the parietal and IFG (including the canonical Broca’s area), during listening to music, and in particular their elicitation through the translation of the typical speech in noise recognition task to a “music emotion in noise recognition task”.

In the Methods section:

“For the analysis of the EEG alpha activity in the left inferior frontal gyrus (including the Broca’s area) and its right homologous were included respectively the channel F7 and F8(55).”

The current results section reads a bit like a laundry list of statistics. It would be helpful to describe the patterns found in the data for each result before presenting and interpreting the relevant statistic. Relatedly, there are several significant behavioral results discussed that do not have an accompanying figure or table to indicate the direction of the effect. Any significant result that is discussed in the manuscript should have a visualization of that result. Results that are not significant do not need a figure, nor do they need the non-significant p-value reported.

We thank the Reviewer for the suggestion, accordingly we modified the text as described in detail below in the response to the points raised by the Reviewer.

The CI sample for the study seems to have some big differences in hearing history and CI use (e.g., prelingual vs. postlingual deafness, word comprehension abilities). I would be interested to see more analyses that test whether these demographic characteristics beyond CI side and duration of use are related to differences seen in alpha in F7/F8.

We thank the Reviewer for the suggestion, accordingly we performed an analysis on the base of the onset of deafness. In particular we performed a t test for comparing pre- and peri-lingual deaf CI users with post-lingual CI users for alpha F8 SNR sub Quiet and for Parietal alpha SNR sub Quiet, since these two indices were the ones that showed an effect of the variable Group. We decided to collapse pre- and peri-lingual deaf persons in the same group because of the numerosity of the groups, in comparison to the post-lingual deaf CI users. This solution allowed to obtain two groups composed by a quite balanced numerosity: pre- + peri-lingual deaf CI users (n=12) and post-lingual-deaf CI users (n=15). Results did not show statistically significant differences, suggesting a lack of influence of the variable onset of deafness on such indices, and therefore on such cognitive task execution. We reported the just mentioned analysis in the text as follows:

In the Result section:

Concerning the potential role of the onset of deafness on the investigated EEG indices, given the effect of the variable Group on F8 alpha SNR sub Quiet and Parietal alpha SNR sub Quiet, it was performed a comparison between a group composed by pre- and peri-lingual deaf CI users and the group composed by post-lingual deaf CI users. Results did not report any statistically significant difference for F8 alpha SNR sub Quiet (t=0.469 p=0.643) neither for Parietal alpha SNR sub Quiet (t=0.710 p=0.484).

In the Discussion section:

The lack of differences within the UCI group on the base of the onset of the deafness in relation to F8 alpha SNR sub Quiet and Parietal alpha SNR sub Quiet indices, support the hypothesis of a lack of influence of such variable on such indices, and therefore on the concurrent cognitive task execution. However, due to the poor numerosity of the two subgroups (pre- and peri-lingual deaf CI users n=12; post-lingual deaf CI users n=15), the result would need further investigations, also eventually expanding the included EEG indices.

In the Method section:

The UCI group was divided into two groups: pre- and peri-lingual deaf CI users and post-lingual deaf CI users. This solution was applied in order to compare these two groups for alpha F8 SNR sub Quiet and for Parietal alpha SNR sub Quiet indices.

The current figures could be improved to be more publication-ready. For example, the caption for Figure 2 references asterisks to indicate significant differences, but they do not appear in the figure. Consider using an alternative to bar graphs that could better indicate the variability in the data (e.g., violin plots), especially given that CI recipients tend to be a very heterogeneous group.

Some unnecessary abbreviations are used for common phrases, which I found unnecessary. For example, using “CR” instead of saying “correct responses.” Using abbreviations only where it improves the flow or in situations where that abbreviation is common (e.g., SNR, UCI, NH) will help with readability.

We thank the Reviewer for the suggestion, accordingly we replaced CR with “correct response”, RT with “Reaction Time” and Q with “Quiet”.

The manuscript needs to be checked for adherence to the journal’s style requirements. For example, in the reference list, I noticed that there are months included in the article references that are not written in English (the date of publication is not actually required for ICJME style for journal articles, the year is sufficient).

Specific comments:

Line 119—“NH group performed better recognition of the emotional content of the musical pieces in comparison to NH group.” –NH group is repeated twice.

We thank the Reviewer for the correction, we corrected the typo as follows:

“NH group performed better recognition of the emotional content of the musical pieces in comparison to UCI”

Line 146—A figure or table should be used to represent the results of difficulty rating.

We thank the Reviewer for the suggestion, and accordingly we added the following figure (new figure 2) to the manuscript:

Figure 3. Left: box plot reporting the percentage of musical excerpts rated for the perceived difficulty: Easy, Medium, Difficult. Right: box plot reporting the percentage of musical excerpts rated for the perceived difficulty (Easy, Medium, Difficult), in combination with the different auditory conditions (Quiet, SNR 10 and SNR5). Whiskers from minimum to maximum value; 25th, 50th (median) and 75th percentile drawn as horizontal lines in each box; “x” stands for the mean.

Line 160—Correlations are reported, but not their significance level. I’d recommend running a correlation test for each of these relationships, and then reporting the p values for the relationships that are significant. The correlation test accounts for not only the strength of the correlation, but also the sample size. (In R, this is done using cor.test() ).

We thank the Reviewer for the correction and added the significance as suggested:

“On all participants, it was found a correlation between the percentage of CR and the percentage of musical pieces rated as Easy (Pearson r=0.525 p=0.004) and Medium to listen (Pearson r=-0.526 p=0.004) (Fig. 6 left), but not for the Difficult to listen musical pieces (Pearson r=-0.246 p=0.207). Finally, in the UCI group it was investigated the correlation between the period of CI use and the rated difficulty, showing a negative correlation with the rated Easy to listen (Pearson r=-0.472 p=0.048) and a positive correlation with the Medium to listen (Pearson r=0.469 p=0.050) musical pieces (Fig. 6 right); no correlation instead was found with the Difficult to listen (Pearson r=0.162 p=0.520) musical pieces.”

Line 167—A figure or table should be used to represent the results of perceived pleasantness.

We thank the Reviewer for the suggestion, and accordingly we added the following figure (new figure 3) to the manuscript:

Figure 4. Left: box plot reporting the percentage of musical excerpts rated for the perceived pleasantness: Liked, Indifferent, Disliked. Right: box plot reporting the percentage of musical excerpts rated for the perceived pleasantness (Liked, Indifferent, Disliked), specifical for the two Groups (UCI and NH). Whiskers from minimum to maximum value; 25th, 50th (median) and 75th percentile drawn as horizontal lines in each box; “x” stands for the mean.

Line 189—I am unsure about the choice to subtract the alpha activity for the Quiet condition from an average of the SNR5 and SNR10 conditions. Please provide a justification here or in the method section for why this was done, as opposed to comparing SNR5 – Quiet and SNR10 – Quiet separately.

We thank the Reviewer for the point raised. We decided to investigate the average of the SNR5 and SNR10 sub Quiet condition because for all F7, F8 and Parietal area alpha activity subtracted of the Quiet condition there was not effect of the different SNR condition (as reported in the text), but there was an effect of the group only after the subtraction of the Quiet condition from the two separate SNRs conditions both for F7 and Parietal area. This suggested us that the presence of background noise, irrespective of the two particular SNRs conditions chosen for the present study, could highlight relevant differences between the two groups. This reasoning was then confirmed by results. However it could be extremely interesting to extend the investigation to further SNRs conditions and assessing whether such evidence could hold true also with more difficult or easy auditory conditions. As suggested by the Reviewer we added the following sentence in the methods section in order to justify the use of the average of the two SNRs sub Quiet:

“In the first part of EEG results it has been showed that for all F7, F8 and Parietal area alpha activity subtracted of the Quiet condition there was no effect of the different SNR condition (SNR5, SNR10 - as reported in the text), but there was an effect of the factor Group only after the subtraction of the Quiet condition from the two separate SNRs conditions both for F7 and Parietal area. This suggested authors that the presence of background noise, irrespective of the two particular SNRs conditions chosen for the present study, could highlight relevant differences between the two groups. This led authors to the choice of investigating the average of the two SNR condition subtracted of the Quiet condition for further ANOVA analyses and correlation analyses with behavioural data.”

Line 210—As I suggested for the behavioral results, please include results of the correlation test with significant p values reported. I suspect that some of these correlations may not be significant—especially the correlation in the third panel of Figure 4, given that there are few participants with long-term CI use.

Line 237—For the discussion section, it would be helpful to first summarize the major behavioral and neural findings before moving to discussion of specific findings, rather than jumping into talking about reaction time right away.

We thank the Reviewer for the valuable suggestion. We added the following sentence at the beginning of the discussion section:

In general, results showed that CI users performed worse than NH controls in the recognition of the emotional content of the stimuli (Fig.1), while reaction times did not appear to be influenced by the factor group (Fig.2). Furthermore, when considering the alpha activity corresponding to the listening to signal to SNR5 sub Q and SNR10 sub Q conditions - ideally removing the emotional content of music and isolating the difficulty level due to the SNRs- CI users reported higher levels of activity in the parietal alpha and in the homologous of the left IFG in the right hemisphere (F8 EEG channel), in comparison to NH (Fig.3). Moreover, the percentage of correct responses, the alpha F8 SNR sub Q and the period of CI use all presented a similar pattern of correlation with the percentage of musical excerpts rated as of Easy and Medium listening difficulty (Fig.4).

Lines 268-272—“Despite the lack of statistically significant correlation between period of CI use and alpha activity, the negative correlations between the period of CI use and the percentage of musical pieces rated as Easy to listen and vice versa the positive correlation with the percentage of musical pieces rated as of Medium difficulty to listen, exactly reflects results obtained through the assessment of the correlation between F8 sub Q and difficulty ratings.” This is a very long sentence, and the point the authors are trying to make is not clear about how these correlations are related.

We thank the Reviewer for the comment. We changed the sentence as follows, hoping in its improvement in clarity:

“Results did not evidence a correlation between period of CI use and alpha activity, however the correlation between the period of CI use and difficulty ratings (negative correlation with musical pieces rated as Easy to listen and positive correlation with musical pieces rated as of Medium difficulty to listen) exactly reflects the correlation between F8 sub Q alpha and difficulty ratings (Fig.4).”

Lines 336-347—With a highly variable group like CI recipients, it would be helpful to include a table to indicate individual characteristics of hearing history and CI use for each person (this could be included as a supplement if desired).

We thank the Reviewer for the useful comment, we added requested information in the annex as follows:

Lines 362-366—It is reported that participants were asked about their music listening habits, but those habits were not considered in the analysis of behavioral or neural data. Consider adding in some analysis of this variable (does frequency of music listening correlate with accuracy of emotion identification, for example?), or remove discussion of the question from the methods.

We thank the Reviewer for the valuable suggestion! We performed and included in the manuscript a correlation analysis between music listening habits and percentage of correct responses, and also between music listening habits and percentage of musical excerpts rated as Easy, Medium and Difficult to listen. However such correlation analyses did not report any statistical significance. In particular, we added in the main text:

“Concerning music listening habits, it was not found any statistically significant correlation between the declared number of hours per week spent listening to music by CI users and percentage of correct responses (Pearson r=0.056 p=0.855), and percentage of musical excerpts rated as Easy (Pearson r=0.155 p=0.614), Medium (Pearson r=-0.240 p=0.429) and Difficult (Pearson r=0.099 p=0.747) to listen.”

Line 374—What was the duration (or average and standard deviation of the duration) of the musical stimuli?

We thank the Reviewer for the comment, the information was already present in the “Stimuli” section of the manuscript:

“From the original database of 32 musical excerpts, after a pre-test conducted on NH university students (n=32) concerning the recognizability of the emotional content of the excerpts, in order to exclude the ones characterized by the poorest recognition, there have been created 2 lists of 24 items, with an average duration of 15.386 ±5.617 seconds per excerpt.”

However we understand that probably we did not exposed it in a clear way, and that this information could be usefully reported also in the part of the manuscript highlighted by the Reviewer, therefore we added it as follows: 

“Each musical stimulus (average duration ± st.d.: 15385 ± 5617 ms) (average duration ± st.d.: 15385 ± 5617 ms)”

Line 420—“EEG recordings were segmented into trials, corresponding to the listening of each musical excerpt.” –This statement is unclear. How long was each of the trial epochs?

We thank the Reviewer for the comment, and apologise for the lack of clarity. In order to analyse the EEG activity in correspondence of the listening to each stimulus, that is each musical excerpt, we segmented the EEG recordings in trials, each corresponding to the exact duration of the musical excerpt, that varied among musical stimuli, as reported in the previous point, with a mean of 15385 ms and ranging from approximately 8000 ms to 28000 ms. Therefore the epoch duration was variable among trials, but it was balanced by the fact that each stimulus was presented under three auditory conditions: Quiet, SNR5 and SNR10. We decided to exclude from the analysis further sections of EEG activity corresponding for instance to the difficulty rating phase, in order to exclude all potential biases and artifacts linked to muscular movements or cognitive processing not strictly related to music fruition under the different auditory conditions. We modified the sentence highlighted by the Reviewer as follows:

“EEG recordings were segmented into trials, corresponding to the listening of each musical excerpt (ranging from approximately 8000 ms to 28000 ms), and excluding all experimental phases, beyond the listening to music, potentially affected by muscular artifacts or cognitive processes for instance linked to the difficulty rating phase.”

Line 248—Provide more explanation about how the sample size calculation was performed.

We thank the Reviewer for the comment, and apologise for the lack of clarity. We added the following sentence to the text in order to explain better the procedures for the sample size calculation:

“In order to perform the sample size analysis, it was employed G*Power 3.1 (102), a free software developed by HHU - Düsseldorf University, selecting as type of power analysis “A priori: Compute required sample size given α, power and effect size, and setting significance level α=0.05, power (1-β)=0.95 and the by default medium effect size f=0.25 (103) (104).”

G*Power represents a free software, developed by HHU - Düsseldorf University, easily retrievable at the following link: https://www.psychologie.hhu.de/arbeitsgruppen/allgemeine-psychologie-und-arbeitspsychologie/gpower.

---

## [Decision Letter · Decision Letter 1]

24 May 2023

PONE-D-22-15819R1Music in noise recognition: an EEG study of listening effort in cochlear implant users and normal hearing controlsPLOS ONE

Dear Dr. Cartocci,

Thank you for submitting your manuscript to PLOS ONE. After careful consideration, we feel that it has merit but does not fully meet PLOS ONE’s publication criteria as it currently stands. Therefore, we invite you to submit a revised version of the manuscript that addresses the points raised during the review process. Most concerns have been addressed. However, the present form of the mansucript needs to be more structured. Please remove all comentaries, ensure that all figures have their axis names and provide a clean version of the manuscript. Clearly separate the text of figure legends from the text of the manuscript.

We look forward to receiving your revised manuscript.

Kind regards,

Paul Hinckley Delano, Ph.D.

Academic Editor

PLOS ONE

Journal Requirements:

Additional Editor Comments:

Most concerns have been addressed. However, the present form of the mansucript needs to be more structured. Please remove all comentaries, ensure that all figures have their axis names and provide a clean version of the manuscript. Clearly separate the text of figure legends from the text of the manuscript.

Reviewers' comments:

Reviewer's Responses to Questions

**Comments to the Author**

1. If the authors have adequately addressed your comments raised in a previous round of review and you feel that this manuscript is now acceptable for publication, you may indicate that here to bypass the “Comments to the Author” section, enter your conflict of interest statement in the “Confidential to Editor” section, and submit your "Accept" recommendation.

Reviewer #1: All comments have been addressed

2. Is the manuscript technically sound, and do the data support the conclusions?

Reviewer #1: Yes

3. Has the statistical analysis been performed appropriately and rigorously? 

Reviewer #1: Yes

4. Have the authors made all data underlying the findings in their manuscript fully available?

Reviewer #1: Yes

5. Is the manuscript presented in an intelligible fashion and written in standard English?

Reviewer #1: Yes

6. Review Comments to the Author

Reviewer #1: I believe the current version of the manuscript to have improved importantly and is now in a condition it can be published

7. PLOS authors have the option to publish the peer review history of their article (what does this mean?). If published, this will include your full peer review and any attached files.

Reviewer #1: **Yes: **Hayo A Breinbauer

---

## [Author Response · Author response to Decision Letter 1]

19 Jun 2023

Dear Editor, we thank for your final comments, please find the response to the following points raised:

Please remove all comentaries, ensure that all figures have their axis names and provide a clean version of the manuscript. 

We apologise for the mistakes. We added axis title wherever it was not already specified and provided a clean version of the manuscript called “Manuscript_v2”

Clearly separate the text of figure legends from the text of the manuscript.

In the just mentioned revised manuscript we added a double space after captions.

Thanks again for your work and useful comments!

---

## [Editor Report · Decision Letter 2]

28 Jun 2023

Music in noise recognition: an EEG study of listening effort in cochlear implant users and normal hearing controls

PONE-D-22-15819R2

Dear Dr. Cartocci,

We’re pleased to inform you that your manuscript has been judged scientifically suitable for publication and will be formally accepted for publication once it meets all outstanding technical requirements.

Kind regards,

Paul Hinckley Delano, Ph.D.

Academic Editor

PLOS ONE
---

## [Editor Report · Acceptance letter]

1 Aug 2023

PONE-D-22-15819R2 

Music in noise recognition: an EEG study of listening effort in cochlear implant users and normal hearing controls 

Dear Dr. Cartocci:

I'm pleased to inform you that your manuscript has been deemed suitable for publication in PLOS ONE. Congratulations! Your manuscript is now with our production department. 

Kind regards, 

on behalf of

Dr. Paul Hinckley Delano 

Academic Editor

PLOS ONE